# Sex-Related Differences in Chronic Myeloid Neoplasms: From the Clinical Observation to the Underlying Biology

**DOI:** 10.3390/ijms22052595

**Published:** 2021-03-05

**Authors:** Theodoros Karantanos, Tania Jain, Alison R. Moliterno, Richard J. Jones, Amy E. DeZern

**Affiliations:** 1Division of Hematological Malignancies and Bone Marrow Transplantation, Sidney Kimmel Comprehensive Cancer Center, Johns Hopkins University, Baltimore, MD 21231, USA; tjain2@jhmi.edu (T.J.); rjjones@jhmi.edu (R.J.J.); Adezern1@jhmi.edu (A.E.D.); 2Division of Adult Hematology, Department of Medicine, Johns Hopkins University, Baltimore, MD 21231, USA; amoliter@jhmi.edu

**Keywords:** myeloid neoplasms, MDS, MPN, sex-related differences

## Abstract

Chronic myeloid neoplasms are clonal diseases with variable clinical course and outcomes and despite the introduction of novel therapies, patients with high-risk disease continue to have overall poor outcomes. Different groups have highlighted that men have overall worse survival and higher incidence of transformation to acute leukemia compared to women across neoplasms such as myelodysplastic syndrome (MDS), myeloproliferative neoplasms (MPN), MDS/MPN overlap neoplasms, and CML. More recent studies evaluating the genomic profile of patients with these neoplasms demonstrated a male predominance for mutations in high-risk genes including *ASXL1*, *U2AF1*, *SRSF2* and *ZRSR2*. The understanding of the underlying biology is limited but a number of hypotheses have been developed and are currently being investigated. This review summarizes the current knowledge about sex-related differences in the clinical outcomes and genomic profile of patients with chronic myeloid neoplasms and discusses the hypothesized biologic mechanisms as an attempt to explain these observations.

## 1. Introduction

Chronic myeloid neoplasms represent a broad spectrum of disorders ranging from myelodysplastic syndrome (MDS) characterized by dysplasia in the marrow and cytopenias in the peripheral blood to myeloproliferative neoplasms (MPN) characterized by hyperplasia in the marrow and elevated counts in the peripheral blood. These diseases can lead to failure of normal hematopoiesis or transformation to acute myeloid leukemia (AML). Despite the variability in their pathology and clinical presentation these neoplasms arise from malignant stem and progenitor cells carrying somatic mutations [1]. The cure of these patients is challenging since malignant stem cells are resistant to chemotherapy and can result in relapse following remission even after allogeneic transplantation [2]. The better understanding of the pathophysiology of chronic myeloid neoplasms is a necessity for the development of new therapeutic approaches to improve the survival of these patients.

It has been well described that men have overall higher incidence of cancer and higher cancer-related mortality [3]. Similarly, chronic myeloid neoplasms and particularly MDS are more common among men [4]. Various reports have highlighted that men with these neoplasms present with more aggressive phenotypes and have worse overall survival and higher incidence of disease progression to AML [5,6]. Moreover, differential response to traditional therapies has been observed between women and men with MDS [7]. Recent data from various cohorts support that these differences are associated with distinct molecular characteristics [8]. However, these sex-related differences have not been taken into consideration in the assessment of the prognosis of patients with chronic myeloid neoplasms and the exact underlying biologic mechanisms remain unknown.

The confirmation of these differences in bigger cohorts and prospective studies may allow the introduction of sex as a prognostic factor in the risk assessment tools used in the clinic. Finally, a deeper understanding of the underlying biology would be particularly valuable as it can provide scientific rationale and potentially lead to the discovery of novel oncogenic pathways implicated in the progression of these neoplasms.

The aim of this review is to summarize the existing literature on sex-related differences in the clinical presentation and outcomes of patients with chronic myeloid neoplasms, discuss the recent advances in the association of these clinical observations with differences at the genomic landscape between women and men and present the current biologic hypotheses attempting to explain these findings.

## 2. Sex-Related Differences in the Presentation and Outcomes of Patients with Chronic Myeloid Neoplasms

Patient characteristics, such as age, performance status and comorbidities, affect the clinical presentation and outcomes of chronic myeloid neoplasms and have been included in risk assessment tools used in the clinic [9,10]. Various reports have highlighted an independent impact of sex in the presentation and outcomes of these patients [5,6,11]. In this section we will summarize the data from a number of MDS, MDS/MPN, MPN, and chronic myeloid leukemia (CML) cohorts supporting a possible implication of sex in the presentation and outcomes of these diseases.

### 2.1. Precursor States (CHIP, ICUS, CCUS)

Clonal hematopoiesis of indeterminate potential (CHIP) is defined as the presence of at least one somatic mutation that is relevant clinically and is otherwise found in MDS (or other myeloid neoplasms) without the presence of persistent cytopenias or diagnosis of myeloid neoplasms [12]. Male sex has been associated with a modestly increased risk of CHIP [13] but there is no strong evidence to support that sex affects significantly the outcomes of these individuals [14]. Idiopathic cytopenia of undetermined significance (ICUS) is defined as relevant cytopenia which is persistent for at least 6 months not explained by another disease and clonal cytopenia of undetermined significance (CCUS) is defined as persistent cytopenia for at least 4 months not explained by another disease with the presence of one or more somatic mutations [15]. Individuals with CCUS show a skewed male-to-female ratio [15] but the impact of sex on the outcomes of individuals with these precursor states has not been extensively evaluated.

### 2.2. MDS

Early data provided evidence that MDS is overall more common in men across various age groups [16] but the role of sex as a prognostic factor for MDS has been only recently highlighted. Nösslinger et al. analyzed 897 MDS patients to evaluate the impact of sex and age in a Cox regression model including R-IPSS score as a variable [17]. They demonstrated that among patients with low and intermediate-1 R-IPSS scores men had significantly worse survival compared to women while there was no significant difference between women and men at higher risk groups [17]. Wang et al. studied the outcomes of 34,681 patients with MDS and showed that male sex is a predictor of worse survival independent of age, race, and sub-type [5]. The authors reported that the negative impact of male sex was significant among patients with refractory anemia, refractory cytopenia with multilineage dysplasia, and MDS with 5q deletion while no significant differences were noted in higher-risk sub-types such as refractory anemia with excess blasts and treatment related MDS [5]. These results indicate an important implication of sex in the outcomes of patients with MDS warranting further evaluation. Based on these data, the introduction of sex as an independent predictor of outcomes in risk assessment tools used in clinical trials and everyday practice is a reasonable consideration.

### 2.3. MDS/MPN

MDS/MPN overlap syndromes is a heterogeneous group of malignancies with overlapping features of both MDS and MPN showing a male predominance which is more prominent in chronic myelomonocytic leukemia (CMML) [18]. Wang et al. highlighted that among 1666 patients with MDS/MPN syndromes men have worse survival compared to women [5]. Our group recently studied retrospectively the outcomes of 167 patients with MDS/MPN and confirmed that men have worse overall survival independent of the specific disease sub-type (Karantanos et al., under review).

### 2.4. MPN

MPN are clonal myeloid neoplasms including essential thrombocytosis (ET), polycythemia vera (PV) and myelofibrosis (MF) sharing common driver mutations in the *JAK2*, *CALR* or *MPL* genes but with a notable variability in the clinical presentations and outcomes [19]. For ET and PV it remains challenging to predict which patients are going to progress to MF and for MF patients the acquisition of additional somatic mutations by itself does not fully explain the variability in the incidence of AML transformation and survival outcomes [20].

Sex is an important factor affecting the presentation of patients with MPN. Women have overall a higher prevalence of MPN, they are younger at diagnosis but they predominate in ET while men predominate in PV and PMF [4,6,21]. Interestingly, women tend to develop worsening symptoms especially abdominal discomfort, headaches, dizziness and fatigue with overall higher total symptoms score [4]. Similarly, women have higher incidence of vascular complications and particularly abdominal venous thrombosis [6,22]. Of note, based on a single center study of 84 consecutive MPN cases with splachnic vein thrombosis, 67% of these patients are women and median age at diagnosis is 54 years supporting that abdominal venous thrombosis is particularly common among young women [22]. Given that younger women have higher estrogen levels and the known impact of estrogens on thrombosis development it is possible that this is the underlying mechanism implicated in these differences. On the contrary, men have higher red blood cell transfusion requirements and worsening thrombocytopenia [4]. Barraco et al. showed that men with secondary MF present not only with lower platelets but also bigger spleens, higher percentage of circulating blasts and higher incidence of complex karyotype [23]. These results support that men tend to have more aggressive MPN phenotypes compared to women who are more symptomatic and carry a higher risk of thrombosis.

The impact of sex in the clinical outcomes of MPN patients has been studied by different groups over the last decade. Tefferi et al. analyzed 1494 patients with ET demonstrating that male sex is associated with worse survival independent of patients’ age, leukocyte count and IPSET score [24]. Our group has also found that men with MPN have worse survival compared to women independent of their age at diagnosis, disease sub-type and driver mutation [6]. Moreover, among ET and PV patients, male sex was associated with a more rapid progression to MF independent of age and disease sub-type at diagnosis [6]. Consistently, men with post-ET and post-PV secondary MF have worse survival independent of their age at the time of disease transformation and their disease sub-type at diagnosis [23]. Finally, a recent retrospective analysis of >2000 individuals with MPN showed that male sex is an independent predictor of higher incidence of transformation to MF and worse overall survival for all disease sub-types [25]. These results support an independent impact of male sex in the outcomes of MPN patients suggesting that adding sex to the risk assessment tools that are used in the clinic to predict the outcomes of MPN patients may be needed particularly for ET and PV patients.

### 2.5. CML

The natural history of CML has been altered significantly following the introduction of tyrosine kinase inhibitors (TKI) with a tremendous improvement of patients’ survival [26]. Given that CML is a molecularly defined neoplasm, significant sex-based alterations in the outcomes would not be expected. However, the review of studies performed in the pre-TKI era reveals interesting sex-related differences in the presentation and outcomes of patients with CML.

Sokal et al. in an early study highlighted that male sex is a negative prognostic indicator independent of spleen size, hemoglobin, levels, platelet counts and percentage of circulating blasts among young (<45 years) patients with CML [27]. Berger et al. analyzed 856 patients with CML and found that women presented with higher platelet counts, and smaller spleen sizes [11]. Moreover, men with CML had a higher incidence of additional chromosomal aberrations and worse survival [11]. This was independent of the risk assessment based on the Sokal score and the difference was more prominent among patients with low and intermediate risk [11]. During the TKI-era, despite that the incidence of TKI switching is higher among women based on data from the SIMPLICITY study [28] there is no strong evidence of sex disparities in the efficacy of TKI with regards to achievement of deep cytogenetic and molecular remissions. Similarly, no sex-related differences in the overall survival of CML patients during the TKI-era have been demonstrated [26,29].

Based on the sex-related differences in the presentation and outcomes of CML patients before the use of TKIs, it could be hypothesized that male sex may be implicated in the acquisition of secondary molecular events driving disease progression.

The observed differences in the presentation and outcomes between women and men with chronic myeloid neoplasms are summarized in Table 1.

## 3. Sex-Related Differences in the Genomic Profile of Patients with Myeloid Neoplasms

Chronic myeloid neoplasms are characterized by the presence of genomic alterations in stem and progenitor cells and the analysis of the genomic landscape of these neoplasms has significantly improved our understanding of their biology and has led to the introduction of novel therapeutic agents for their treatment [1,19]. Despite the epidemiologic evidence supporting that men with chronic myeloid neoplasms have worse outcomes compared to women the evaluation of these differences at the genomic level has only recently been initiated.

The comparison of the frequency of specific CHIP-related somatic mutations between women and men has revealed a modest increase in the frequency of *DNMT3A* and *TET2* mutations in women with CHIP [30]. De-Morgan et al. studied the genomic data of 2773 MDS patients and showed that men have higher incidence of *U2AF1* and *ZSRS2* mutations whereas women with MDS have higher incidence of *DNMT3A* and *TP53* mutations [8]. Bose et al., performed an analysis of the genomic landscape of 102 patients with unclassifiable MDS/MPN syndrome and further confirmed that women have a higher incidence of *DNMT3A* mutations while mutations in the *ASXL1* gene were more common among men [31]. We recently studied the mutational landscape of 100 patients with MDS/MPN and found that men have a higher number of somatic mutations and higher frequency of mutations in the *EZH2* gene independent of the specific disease sub-type (Karantanos et al., under review).

The analysis of the next generation sequencing data of 227 patients with MPN revealed that men have a higher number of non-MPN-specific somatic mutations and higher incidence of one or two mutations in genes such as *ASXL1*, *IDH1/2*, *U2AF1*, *SRSF2* and *EZH2* [6]. These genes have been associated with worse outcomes in MPN patients especially when two of these mutations co-occur [32].

Despite that the connection between the sex-related differences in the mutational profile and the worse clinical outcomes of men is still missing, these results suggest that men with chronic myeloid neoplasms have overall more mutations in genes involved in RNA splicing and epigenetic regulation and confer a higher risk for disease progression and overall worse outcomes.

The evaluation of AML cohorts has also revealed interesting findings associated with sex-related differences that can be utilized to extract useful conclusions given that a significant percentage of AMLs are derived from underlying chronic myeloid neoplasms. De-Morgan et al. demonstrated that men with AML had more often “pre-leukemic” somatic mutations which are found usually in chronic myeloid neoplasms such as mutations in *ASXL1*, *U2AF1*, *SRSF2*, *BCOR* and *RUNX1* genes [8]. The higher incidence of “pre-leukemic” mutations in men with AML was also highlighted by Metzeler et al. who showed that mutations in *RUNX1*, *ASXL1*, *SRSF2*, *STAG2*, and *BCOR* genes were more prevalent in men compared to women in a cohort of 664 AML patients [33]. Finally, Engen et al. reported in a preprint article that men with FLT3-ITD mutated AML have higher incidence of *RUNX1*, *ZRSR2*, *SRSF2*, *U2AF1*, *ASXL1* and *EZH2* mutations [34]. These results indicate a male predilection for mutations that have been reported to drive the progression of chronic myeloid neoplasms and their transformation to AML. Thus, these conclusions could provide a rationale for the worse outcomes of men with these neoplasms.

## 4. Hypothesized Biologic Mechanisms Implicated in the Gender-Related Differences in Myeloid Neoplasms

The underlying biologic mechanisms implicated in the sex-related differences in the clinical outcomes and genomic profile of patients with chronic myeloid neoplasms have not been well studied. Men in the USA carry an age-adjusted excess risk of 20.4% of developing any cancer and there is ≥2:1 male predominance for some cancer types [35]. These differences have been attributed to factors such as higher prevalence of smoking in men [36]. However, multivariable analysis has revealed that the differences remain significant after adjusting for differences in gross domestic product, geographical region, and environmental risk factors, including tobacco exposure [35,37]. Moreover, despite the declining smoking rates among men, the male predominance in several cancers such as renal cell, bladder and head and neck cancers have remained >2:1 [3]. Thus, it is possible that other biologic characteristics differing between genders could be involved in the development of more aggressive phenotypes in various cancers including myeloid neoplasms.

### 4.1. EXITS Hypothesis

The “escape from X-inactivation tumor-suppressor” (EXITS) hypothesis is that biallelic expression of genes encoding tumor suppressors located in the non-pseudoautosomal region (PAR) of X-chromosome affords females enhanced cancer protection, which substantively contributes to the observed higher incidence of some tumors in men [38,39,40]. Dunford et al., reported that six out of 783 of non-PAR X-chromosome genes (*ATRX*, *CNKSR2*, *DDX3X*, *KDM5C*, *KDM6A*, and *MAGEC3*) harbor more often loss-of-function mutations in men compared to women across 21 different cancer types [41] providing further evidence supporting the EXITS hypothesis. A subset of genes reported to be more frequently mutated among men with AML such as *BCOR*, *PHF6*, *STAG2* and *ZRSR2* are encoded by chromosome X [8]. Thus, an oncogenic allele will always be expressed from the single copy of the X chromosome in men while the same allele carried on the inactive X chromosome will have no significant impact if the gene does not escape X-chromosome inactivation [8]. Consistently, Yoshida et al. demonstrated that nonsense or frameshift mutations in *ZRSR2* causing either a premature truncation or a large structural change of the protein, leading to loss-of-function are found exclusively among men and not women [42]. More recently in a pre-print, Togami et al. reported that loss-of-function *ZRSR2* mutations are enriched in blastic plasmacytoid dendritic cell neoplasm, a rare leukemia with a three times higher incidence among men and almost all these mutations occur in men [43].

The initiation and maintenance of the X-chromosome inactivation requires *Xist*, which is a non-coding X-linked gene and is expressed only in females. Of note, the silencing of this gene causes aberrant maturation and age-dependent-loss in hematopoietic stem cells through X chromosome re-activation. Particularly, *Xist* deletion in the hematopoietic compartment of mice led to the development of a highly aggressive MDS/MPN phenotype in female mice.

Overall, these results support that potentially X-chromosome inactivation can protect women from loss-of-function mutations in genes that are implicated in the development and progression of chronic myeloid neoplasms and particularly genes involved in splicing machinery and transcriptional regulation.

### 4.2. Sex-Related Differences in the Metabolism of DNA Methyltransferase Inhibitors

An alternative hypothesis to explain the different outcomes between women and men with myeloid neoplasms is that they could have different responses to the available therapies. Particularly, the cytidine analogue drugs azacitidine and decitabine are the most commonly used FDA approved agents for the treatment of patients with MDS and MDS/MPN [44]. These agents induce the apoptosis and differentiation of malignant cells through the alteration of their epigenetic profile and depletion of DNA methyl-transferase 1 (DNMT1) after incorporation into DNA [45]. These drugs are rapidly deaminated to uracil base moiety counterparts by the ubiquitously expressed enzyme cytidine deaminase (CDA) [46] leading to particularly low half-lives in vivo [47].

Mahfouz et al. found that men have significantly higher CDA activity compared to women associated with worse overall survival in men with MDS treated with azacitidine or decitabine compared to women [48]. The authors studied the pharmacokinetics and pharmacodynamics of the cytidine analogues in mice and showed that decitabine clearance is more rapid in male mice which was associated with higher liver CDA expression and more prominent depletion of DNMT1 in their bone marrow compared to female mice [48]. A mathematical analysis based on the concept that the efficacy of these drugs is higher during the S-phase of the cell cycle suggested that this male sex-related decrease in the half-lives of cytidine analogues would produce a substantially greater decrease in their efficacy in neoplasms with a lower percentage of malignant cells in the S-phase of the cell cycle such as MDS as opposed to AML [48].

Possible implications of these differences in the clinic have already been studied. DeZern et al. evaluated the impact of sex on the response of MDS patients to cytidine analogues and found that women treated with decitabine had significantly better overall survival compared to women treated with azacitdine while treatment with decitabine or azacitidine did not affect the survival of men [7]. Given the findings by Mahfouz et al. supporting that men have a higher CDA activity, the authors hypothesized that this difference could be related to a more rapid inactivation of azacitidine compared to decitabine in women proposing that a sex-specific dose adjustment of these agents could be considered.

Finally, another important implication of sex-related differences in the metabolism of cytidine analogues is the assessment of response of men and women to oral cedazuridine/decitabine, which was recently approved for patients with MDS and CMML [49]. Cedazuridine is a CDA inhibitor and the evaluation of the impact of sex in the response to this combination would be of particular interest.

### 4.3. Sex-Related Differences in the Disease Burden in the Primitive Cells’ Compartment

The increased burden of somatic mutations in bone marrow samples has been associated with overall worse survival in patients with myeloid neoplasms [50,51,52,53]. Higher allele burden of somatic mutations in the CD34+ cells has been associated with overall more aggressive MDS sub-types [54] and congruence of the *JAK2V617F* CD34+ progenitor and neutrophil allele burdens in MPN patients has been correlated with more advanced disease and MF phenotype [55]. Our group analyzed the *JAK2V617F* allele burden in the neutrophils of 524 MPN patients and found that there are no significant differences between men and women across the different MPN phenotypes [6]. However, we did notice that men with PV have a trend toward higher *JAK2V617F* allele burden compared to women with PV [6] which is consistent with a previous analysis [56] while among patients with primary MF women had higher *JAK2V617F* allele burden in their neutrophils [6]. However, when we studied the *JAK2V617F* allele burden in the CD34+ cells of 121 patients we found that men have higher allele burden independent of the specific MPN phenotype with the difference being more prominent in the lower-risk phenotypes of ET and PV [6].

More rapid expansion of neoplastic clones in the stem and progenitor compartments in men could be an interesting alternative hypothesis providing a rational for their overall worse outcomes and faster progression to higher-risk phenotypes and acute leukemia. However, further studies in bigger cohorts of different myeloid neoplasms should be performed to confirm these findings. Finally, if this observation is confirmed the underlying molecular biology needs to be further elucidated.

## 5. A Potential Role of Hormonal Receptors

Estrogen and androgen receptors (ER and AR, respectively) have been implicated in the development and progression of malignancies such as breast, prostate and gastric cancer [57]. Numerous studies have highlighted the involvement of hormonal receptors in various oncogenic cellular functions such as cell cycle progression [58], DNA damage repair [59], and cytokine signaling regulation [60]. However, it is unclear if hormonal receptor signaling is associated with the progression of myeloid neoplasms.

Sanchez-Aguilera et al. showed that the ER induces the apoptosis of *JAK2V617F* mutated hematopoietic stem cells and its activation by tamoxifen suppresses the progression of *JAK2V617F*-mediated MPN disease in mice and increases the sensitivity of MLL-AF9+ leukemias to chemotherapy [61]. These findings support that ER activation could suppress the growth of malignant stem and progenitor cells in the bone marrow providing an explanation for the lower *JAK2V617F* allele burden in the CD34+ cells of women [6]. However, the higher incidence of myeloid neoplasms in men and their worse outcomes appear to be independent of age or they are more prominent in older individuals [5,17]. Given that estrogen levels are significantly decreased in post-menopausal women, the sex-related differences would be expected to be less prominent in older patients. Finally, as most of the myeloid neoplasms occur in elderly individuals, it remains unclear if ER signaling has an important role in the alteration of the pathogenesis of these diseases.

Based on a study by Chuang et al., AR knockout in a transgenic mouse model leads to the development of significant neutropenia due to suppression of terminal differentiation of granulocytes [62]. The authors showed that AR increases the sensitivity of myelocytes to granulocyte colony stimulating factor (GCSF) and regulates the expression of various GCSF-target genes [62]. Further understanding of the implication of AR signaling and evaluation of a possible role of AR in the development and progression of myeloid neoplasms would be interesting as it can provide another hypothesis to explain the sex-related differences observed in patients with myeloid neoplasms and can be easily translated to novel therapeutic approaches.

## 6. Conclusions

Based on data from different cohorts, men with myeloid diseases have overall worse outcomes compared to women, which tend to be more prominent among patients with lower risk phenotypes. These results have been associated with reproducible sex-related differences in the genomic landscape of patients with myeloid neoplasms demonstrating that men have higher incidence of mutations in high-risk genes such as *ZRSR2*, *U2AF1*, *SRSF2*, and *ASXL1*. Despite that various hypotheses have been developed including the X-chromosome inactivation hypothesis, the different metabolism of cytidine analogues, alterations in the primitive cells’ compartment and implication of hormonal receptors, these differences remain not well understood. It is possible that more than one of these mechanisms contribute to these clinical observations. Better understanding of the underlying biology can improve the prognostication of patients with myeloid neoplasms and provide opportunities for exciting novel therapies. Finally, further confirmation of these differences could permit the addition of sex as an independent prognostic factor in the risk-assessment tools used in the clinic especially for patients with lower risk diseases.

## Figures and Tables

**Table 1 ijms-22-02595-t001:** Summary of the observed differences in the presentation and outcomes between women and men with chronic myeloid neoplasms.

Myeloid Neoplasm	Observation	Reference
MDS	The frequency is higher in men compared to women	[16]
MDS—Low and Intermediate-1 R-IPSS score	Men have worse survival compared to women	[17]
MDS—RA, RCMD, 5q	Men have worse survival compared to women	[5]
MDS/MPN	Men have worse survival compared to women	[5]
MPN	Women predominate in ET and men predominate in PV and PMF	[6,21]
MPN	Women have higher incidence of venous thrombosis	[6,22]
MPN	Men have lower platelets	[4,23]
PMF	Men have bigger spleens, higher percentage of circulating blasts and higher incidence of complex karyotypes	[23]
ET	Male sex is an independent predictor of worse survival	[24]
ET, PV	Male sex is an independent predictor of worse survival and higher incidence of transformation to MF	[6,25]
Secondary MF	Male sex is an independent predictor of worse survival	[23]
MPN	Men have worse survival across all the subtypes	[6,25]
CML	Male sex is an independent predictor of worse survival among young patients (<45 years old)	[27]
CML	Men have bigger spleens, lower platelets, higher incidence of additional chromosomal abnormalities and worse survival among patients with low and intermediate risk groups	[11]

**Abbreviations:** MDS, myelodysplastic syndrome; R-IPSS, revised international prognostic scoring system; RA, refractory anemia; RCMD, refractory cytopenia with multilineage dysplasia; MDS/MPN, myelodysplastic/myeloproliferative overlap neoplasm; MPN, myeloproliferative neoplasm; ET, essential thrombocythemia; PV, polycythemia vera; PMF, primary myelofibrosis; MF, myelofibrosis; CML, chronic myeloid leukemia.

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
