# Peer review of "Sex-Related Differences in Chronic Myeloid Neoplasms: From the Clinical Observation to the Underlying Biology"

_ijms, 2021, doi:10.3390/ijms22052595_

Round 1

Reviewer 1 Report

Dr. Theodoros Karantanos et al. introduced Sex-related differences in chronic myeloid neoplasms, including phenotypes, mutations, and potential mechanisms. Overall, the manuscript was organized well, and the interpretation is in details. I only have some minor issues below:

  1. The English need improvement.
  2. Line 239, ““escape from X-inactivation tumor-suppressor” (EXIST)”, the initial is “EXITS”, not “EXIST”.
  3. The authors mentioned “X-inactivation”, I think authors should also introduce the lnc RNA XIST, which is a long non-coding RNA that can lead to “X-inactivation” and itself serves as a tumor suppressor, only expressed in the female.

Author Response

We would like to thank the reviewer for his positive remarks and his very interesting comments. Please see the point-by-point response to the reviewer's comments below:

  1. The English need improvement

We thank the reviewer for his comment and as a response we performed extensive review and editing of the English of our manuscript

2.  Line 239, ““escape from X-inactivation tumor-suppressor” (EXIST)”, the initial is “EXITS”, not “EXIST”.

We appreciate the reviewer's comment and as a response we corrected the term in our revised manuscript.

3.  The authors mentioned “X-inactivation”, I think authors should also introduce the lnc RNA XIST, which is a long non-coding RNA that can lead to “X-inactivation” and itself serves as a tumor suppressor, only expressed in the female.

We agree with the reviewer's point and as a response we introduced the XIST gene describing its potential role as a tumor suppressor in hematologic malignancies.

Reviewer 2 Report

This is an interesting review of sex differences in MDS and MPNs and a look at the what is known about the underlying biology

I have some minor comments

 In line 135 it is briefly referred to vascular events and some predominance in females. However, in clinical practice the predominance of particularly intraabdominal thrombosis in young females is very notable Should this theme be developed more here and the issue of hormones included?

line 79 Patient characteristics would be acceptable (thus avoiding the dreaded apostrophe)

line 90-92 : would some figures from the references be useful to support the terms modest and strong

Line 92/93: The terms ICUS and CCUS need to be in full when first used

line 286 mathematical (not mathematic)

Author Response

We would like to thank the reviewer for his positive comments. We have included the point-by-point response to his comments below:

  1. In line 135 it is briefly referred to vascular events and some predominance in females. However, in clinical practice the predominance of particularly intraabdominal thrombosis in young females is very notable Should this theme be developed more here and the issue of hormones included?

We agree with the reviewer's comment. As a response we have added a reference describing that 2/3 patients with splachnic vein thrombosis are women with median age of 54 years supporting a possible role of estrogens in these differences.

line 79 Patient characteristics would be acceptable (thus avoiding the dreaded apostrophe)

We thank the reviewer for this note. We deleted the apostrophe and changed the term to patient characteristics

line 90-92 : would some figures from the references be useful to support the terms modest and strong

We appreciate the reviewer's comment and we definitely agree that this would provide some understanding of the terms modest and strong but unfortunately, the original reports did not include any relevant figures.

Line 92/93: The terms ICUS and CCUS need to be in full when first used

We thank the reviewer for this comment. We have added the full terms of these two conditions before abbreviating them

line 286 mathematical (not mathematic)

We agree with the reviewer. We have changed the term to mathematical.